# In Silico Genome-Wide Characterisation of the Lipid Transfer Protein Multigenic Family in Sunflower (*H. annuus* L.)

**DOI:** 10.3390/plants11050664

**Published:** 2022-02-28

**Authors:** Alberto Vangelisti, Samuel Simoni, Gabriele Usai, Flavia Mascagni, Maria Ventimiglia, Lucia Natali, Andrea Cavallini, Tommaso Giordani

**Affiliations:** Department of Agriculture, Food and Environment, University of Pisa, Via del Borghetto 80, 56124 Pisa, Italy; samuel.simoni@phd.unipi.it (S.S.); gabriele.usai@agr.unipi.it (G.U.); flavia.mascagni@unipi.it (F.M.); maria.ventimiglia@gmail.com (M.V.); lucia.natali@unipi.it (L.N.); andrea.cavallini@unipi.it (A.C.); tommaso.giordani@unipi.it (T.G.)

**Keywords:** lipid transfer proteins (LTPs), sunflower, genome-wide, multigenic family

## Abstract

The sunflower (*Helianthus annuus* L.) is among the most widely cultivated crops in the world due to the oilseed production. Lipid transfer proteins (LTPs) are low molecular mass proteins encoded by a broad multigenic family in higher plants, showing a vast range of functions; these proteins have not been characterised in sunflower at the genomic level. In this work, we exploited the reliable genome sequence of sunflower to identify and characterise the LTP multigenic family in *H. annuus*. Overall, 101 sunflower putative LTP genes were identified using a homology search and the HMM algorithm. The selected sequences were characterised through phylogenetic analysis, exon–intron organisation, and protein structural motifs. Sunflower LTPs were subdivided into four clades, reflecting their genomic and structural organisation. This gene family was further investigated by analysing the possible duplication origin of genes, which showed the prevalence of tandem and whole genome duplication events, a result that is in line with polyploidisation events that occurred during sunflower genome evolution. Furthermore, LTP gene expression was evaluated on cDNA libraries constructed on six sunflower tissues (leaf, root, ligule, seed, stamen, and pistil) and from roots treated with stimuli mimicking biotic and abiotic stress. Genes encoding LTPs belonging to three out of four clades responded specifically to external stimuli, especially to abscisic acid, auxin, and the saline environment. Interestingly, genes encoding proteins belonging to one clade were expressed exclusively in sunflower seeds. This work is a first attempt of genome-wide identification and characterisation of the LTP multigenic family in a plant species.

## 1. Introduction

Sunflower (*Helianthus annuus* L.) is an annual plant belonging to the Asteraceae, the largest family of the Angiosperm clades. It is an important source of vegetable oil and is extensively cultivated worldwide. Historically, *H. annuus* was initially cultivated in North America, where it was domesticated by Native Americans before being exported to Europe; after breeding, it became a globally important crop [1].

Because of its importance as an oilseed crop, lipid metabolism and accumulation in sunflower seeds have been largely studied [2,3]. More than 400 genes involved in oil metabolism were detected in the sunflower genome, which encodes for proteins implied in metabolic pathways leading to 160 different oil metabolites [4]. In addition, as in other plant species, lipids play important roles in growth and development, building and maintaining energy stores and membranes necessary for the compartmentalisation of different metabolic pathways in the cell [5]. Furthermore, lipids mediate responses to stress by being involved in cell signalling, constructing the surface cuticle layer, and protecting cells from water stress-related desiccation. Finally, membrane lipids mediate cell signalling associated with responses to the environment [5].

Lipid transfer proteins (LTPs) can bind a variety of hydrophobic molecules, such as fatty acids, fatty acyl-CoA, phospholipids, glycolipids, and cutin monomers [6]. They are encoded by a multigenic family in higher plants and are characterised by a generally low molecular mass spanning from 7 to a maximum of 10 KDa [7]. Since their discovery (about 40 years ago), the list of proteins belonging to this family has expanded, especially in seed plants [8]. However, the classification of this family has often been simplistically reduced to their molecular size [8]. In fact, LTPs have been classified by analysing molecular mass data; for example, in *Arabidopsis thaliana*, two groups of 7 and 9–10 KDa were distinguished [6,9]. Another frequently used classification is based on the occurrence of conserved cysteine domains with the general form C-Xn-C-Xn-CC-Xn-CXC-Xn-C-Xn-C [10]. However, both classifications consider only partially genetic features, being based exclusively on biochemical properties and not considering the genetic structure of LTPs. In addition, these methods can more accurately predict the so-called non-specific LTPs [6,9], which are implied principally in a biological role in the plant cell cytoplasm (e.g., signalling, and defence), excluding other possible members of this family that could have been generated by duplication and subsequent divergence, a common occurrence in complex genomes, such as that of sunflower. Thus, a comprehensive classification for LTPs still does not exist, and possible clustering of these proteins could change for different plant species [8].

Lipid transfer proteins have been localised by using biochemical and immunolocalization approaches specifically in the apoplast of several plant species, even though many LTPs were also detected in the intracellular position, in particular in the glyoxysome matrix and storage vacuoles of seeds, such as in the case of *Ricinus communis*, *Vigna unguiculata*, and *Helianthus annuus* [7]. More recently the expression of members of the LTP family has been also detected in different vegetative tissue, especially in leaves, stem, and seedlings as described for barley, rice, and *Arabidopsis thaliana* [10]. In addition, the regulation of LTP of type 1 and 2 has been observed during the development of anther and microspores in *Arabidopsis thaliana* [11,12].

Many possible roles have been proposed for LTPs, including wax and cutin formation [13], antimicrobial defence [14], signalling [15], receptors for defence response [16], secretion or deposition of extracellular lipophilic material [17], and mobilisation of seed storage lipids [18]. This vast range of functions could derive from the occurrence of multiple LTPs that are encoded by a multigenic family, whose members share partial sequence similarity, encoding proteins possibly involved in distinct biological processes [7].

Recently, the sunflower genome was completely deciphered using third-generation sequencing technology (PacBio), which exploits longer sequences, allowing the assembly of reliable genome sequences [4]. In this context, a total of 52,232 genes and 5803 spliced long non-coding RNAs were discovered in the 3.6 Gbp large genome [4]; in addition, the high abundance of the repetitive DNA of *H. annuus* was confirmed, with about 80% repetitive elements, composed mainly by *Gypsy* and *Copia* retrotransposons, as already observed in previous studies [19,20,21]. The availability of a reliable genome sequence also allowed the study of crypto-polyploidisation events within Asteraceae genome evolution, which occurred over millions of years and passed through two major events of palaeopolyploidization [4,22].

Only a few members of the LTP family have been retrieved and analysed in sunflower; for example, the protein named Ha-AP10 was identified in the seed where it has been associated with possible roles in signalling events during seed germination [7]. Five copies of putative non-specific LTP genes were identified in a single large bacterial artificial chromosome (BAC) sequence of sunflower [23,24].

Considering the importance of sunflower as an oilseed crop and the implication of LTPs in several biological aspects of plant development and adaptation, the aim of this study was to identify and classify all members of the LTP multigenic family in *H. annuus*, exploiting the available, accurate genome sequence of this species. We performed a genome-wide characterisation of LTP sequences at structural and functional levels in several plant organs and during treatments, simulating biotic and abiotic stress. Our work represents one of the first attempts to classify at genomic level this multigenic family in a plant species and is the basis for future studies aimed at establishing the function of different members of the LTP family in plants.

## 2. Results

### 2.1. Identification of LTPs in the H. annuus Proteome

The amino acid sequences of LTPs previously identified in the sunflower genome [23] were used to retrieve candidate LTPs from the *H. annuus* proteome [4]; in the first step, 22 sequences were detected using BLASTP. Then, to retrieve more candidate proteins, the PFAM database was used to find common motifs among these sequences. The only motif found was PF00234 “Protease inhibitor/seed storage/LTP family”. A hidden Markov model approach was used on PF00234 alignment to extend the search to other putative divergent LTP sequences (see Materials and Methods). Following this procedure, we retrieved a total of 101 candidate LTP-encoding genes of *H. annuus*. Among these candidate genes, 75 were unique, 8 were probable pseudogenes, and 8 genes had two or more copies. Moreover LOC110877546, LOC110884099, LOC110908581, LOC110912707, LOC110922502, LOC110930696, and LOC110937439 had two isoforms, and LOC110921897 had four isoforms (Appendix A).

### 2.2. Structural Characterisation of H. annuus LTPs

On average, putative *H. annuus* LTPs (HaLTPs) were made up of 155 amino acids (aa), ranging from 88 to 408 aa (Appendix A). 

To examine the structure and relationships among putative HaLTPs, a phylogenetic analysis based on the occurrence of structural motifs was performed to identify possible gene clusters. Overall, we were able to classify putative LTPs in four groups (Figure 1). The first group was characterised by the presence of at least two of three motifs (Motif1, Motif2, Motif3). The second and third groups were both marked by the occurrence of one motif (Motif5 or Motif4, respectively), and the last group was characterised by the presence of two motifs (Motif2 and Motif4) (Figure 1a, Appendix A), even if in some cases Motif2 and/or Motif4 were not highly conserved (indicated as Motif2 and Motif4, Figure 1a). For three HaLTPs, it was impossible to retrieve any motif (Figure 1a). The consensus sequences of each of the five domains are reported in Appendix A.

Figure 1 reports the exon–intron organisation of genes encoding identified HaLTPs (Figure 1b). In particular, Group 1 of the HaLTP genes was the most uniform and showed a gene structure formed by two exons and one intron, on average, except for an LTP gene that displayed a total of seven exons and one that had three exons. Groups 2 and 3 had one to three exons. Finally, Group 4 was highly variable, displaying one to four exons (Figure 1b). 

The four groups of putative HaLTPs were also characterised by calculating the molecular weight and isoelectric point of the putative mature protein. Regarding the calculated molecular weight, Group 1 showed the closest range of values, ranging from 9 to 15 KDa, similar to Group 3 (average 11 KDa); Group 4 was slightly different, with an average of 15 KDa. Conversely, Group 2 had the highest variability, ranging from 11 to 44 KDa (Figure 2). 

Concerning the calculated isoelectric point, proteins belonging to Groups 1 and 2 were mostly basic, with an average pH of 8.5 and 7.5, respectively; Groups 3 and 4 covered a vast range of pH values from 3.6 to 10.7 (Figure 2).

Considering molecular weight, Group 1 may belong to non-specific LTPs of types 1 and 2. To classify Groups 2, 3, and 4, we searched for sequence similarities on the Swiss-Prot database. In particular, Group 2 HaLTPs showed similarity to 2S seed storage proteins, whereas Groups 3 and 4 shared sequence similarity with DIR1 and LTP G-anchored-5 proteins, respectively (Appendix A).

### 2.3. Chromosomal Localisation and Duplication Events of LTPs Encoding Genes in the H. annus Genome

All putative HaLTP genes were localised on the 17 chromosomes of *H. annuus*. LTPs were distributed over all the sunflower chromosomes, although with different frequencies (Figure 3, Appendix A). Chromosomes 11 and 15 bear the highest number of HaLTP genes, with 17 and 14 genes, respectively (Figure 3).

Putative duplication events involving the HaLTP gene family in the *H. annuus* genome were investigated. Two HaLTP genes located within the frame of 200 kbp were considered duplicated in tandem. Following this approach, we identified 11 pairs of HaLTPs for Group 1, 3 for Group 2, 1 in Group 3, and 4 in Group 4 (Figure 1c). 

Furthermore, we observed segmental duplication events for nine HaLTP pairs (XP_021973854.1–XP_022006327.1; XP_021981388.1–XP_022009213.1; XP_021984981.2–XP_021992525.1; XP_021992525.1–XP_022035551.1; XP_021993622.1–XP_022013239.1; XP_022000196.1–XP_022022835.1; XP_022002419.1–XP_022029738.1; XP_022002421.1–XP_022029740.1; XP_022010101.1–XP_022034230.1) (Figure 1c). These results indicated that tandem and segmental duplication contributed to the diversity and expansion of the LTPs gene family in sunflower. 

The non-synonymous/synonymous ratio was used to consider possible evolutionary constraints. Putative HaLTPs mainly underwent purifying selection, showing a ratio between kn/ks lower then 1 (Appendix A).

Putative HaLTPs genes were also analysed to identify putative pseudogenes. Overall, 8 of the 101 HaLTPs might be pseudogenes (Appendix A). 

### 2.4. Analysis of Putative Cis-Regulatory Elements in HaLTP-Encoding Genes 

Cis-acting elements located within 1500 bp upstream of HaLTP coding sequences were retrieved. The abscisic acid (ABA) and methyl jasmonate (MeJA) responsive elements were the most abundant cis-regulatory elements in each HaLTP group, ranging from 23 for Group 1 to 8 in Group 3 (Figure 4). In contrast, the gibberellic acid (GA) responsive elements had the lowest frequency in each HaLTP group, ranging from 9 to 2 genes (Figure 4).

Other abundant putative cis-acting elements were elements responsive to ethylene in Group 1, to ethylene and auxin in Group 2, and to drought in Groups 3 and 4.

### 2.5. Expression Analysis of the HaLTP Gene Family

Expression of sunflower LTP-encoding genes was analysed in six organs (leaf, root, ligule, pistil, seed, stamen) and in roots of plantlets subjected to various treatments: abscisic acid (ABA), methyl jasmonate (MeJA), ethylene (ACC), brassinosteroids (BRA), gibberellic acid (GA3), auxin (IAA), kinetin (KIN), salt (NaCl), polyethylene glycol (PEG), salicylic acid (SA), and strigolactones (STRI). Group 1 genes showed the highest expression values in the leaf and stamen (average RPKM 828.54 and 1400.7, respectively). The expression of Group 2 genes was high in the seed (average RPKM 9157.43) and very low in the other organs; Group 3 genes had higher RPKM values in the seed and roots (average RPKM 6123.23 and 527.76, respectively) compared to the other organs. Group 4 genes showed lower expression values compared to the other groups; nevertheless, the highest RPKM values were detected in the leaf (average RPKM 44.94) and pistil (average RPKM 35.14; Figure 5).

Finally, cDNA libraries of roots of plantlets subjected to different treatments were compared to respective control plantlets to identify differentially expressed LTP genes (DEGs), considering only HaLTPs with an RPKM > 1 (60 of 101). Significant activation of genes encoding Group 4 LTPs (19 DEGs for ABA, 5 DEGs for IAA, 4 DEGs for SA, 3 DEGs for NaCL, and 1 DEG for ACC) were identified, followed by genes encoding Group 1 (3 DEGs for ABA, 2 DEGs for MeJA, and 1 DEG for IAA) and three proteins (5 DEGS for IAA, 1 DEG for ABA); no differential expression was found for genes of Group 2 LTPs (Figure 6). The complete list of HaLTPs differentially expressed in sunflower roots exposed to biotic/abiotic stimuli is shown in Appendix A.

## 3. Discussion

In this study, 101 putative LTP-encoding genes of *H. annuus* were identified, a number comparable to that observed in other plant species [10,25], considering the complexity of the sunflower genome. These genes were characterised and subdivided into groups using phylogenetic and structural analyses (Figure 1), providing a robust and reliable characterisation of this multigenic family and considering possible divergent gene sequences that originated during sunflower genome evolution. Protein sequence comparisons allowed us to subdivide the family into 4 groups, based on the presence of five motifs in the protein sequences (Figure 1). 

Lipid transfer protein sequences in the sunflower genome that mainly correspond to Group 1 could be associated with *non-specific* LTPs of types 1 and 2 [9], considering the range of molecular weight shown by this group (Figure 2), whereas HMMER [26] was able to retrieve the most divergent members of the LTP multigenic family in *H. annuus* (Groups 2, 3, 4, and partially Group 1).

As for genomic localisation, putative HaLTP genes were widespread on all the sunflower chromosomes, although at different frequencies. Overall, nineteen of 101 sequences were retrieved in regions with low gene density. In many cases, HaLTP gene copies were grouped in chromosome regions (Figure 3). Actually, many HaLTP genes had a possible origin given by duplication, namely by whole genome duplication and tandem duplication events (Figure 1), indicating the proliferation of this family along with sunflower genome evolution. The genome of Asteraceae, including *H. annuus*, evolved in a million-year timespan through multiple polyploidisation events, creating patterns of duplicated genes and chromosome synteny [4,22]. Our results agree with what is known about Asteraceae genome duplications [4]. However, our data indicate that many HaLTP copies originated through tandem duplications of small genome fragments. Some of these genes were possibly subjected to mutations and subsequent inactivation, resulting in the formation of pseudogenes. However, a low number of putative HaLTP pseudogenes were identified in the sunflower genome. 

By measuring the Kn/Ks ratio between duplicated HaLTP gene copies, we observed that, in many cases, duplicated HaLTPs genes underwent purifying selection. Conservation of duplicated sequences, which showed minor amino-acidic substitution among HaLTPs as suggested by the Kn/Ks ratio, might underpin a preserved function for this multigenic family, explaining sequence conservation during sunflower genome evolution (Appendix A).

Concerning regulatory sequence diversification, eight cis-regulatory motifs related to different stimuli mimicking abiotic and biotic stress were detected in each of the 101 putative HaLTP genes. For each HaLTP group, the analysed motifs were detected in at least one member of the group, suggesting that each HaLTP group can be regulated by different stimuli, although some motifs resulted more frequently in certain groups than in others. For example, drought-responsive elements were highly frequent in Groups 3 and 4 and were less frequently represented in Groups 1 and 2 (Figure 4).

In the last decade, sunflower has been the object of many transcriptomic studies intended to analyse gene expression in different organs of the plant and in response to biotic and abiotic stress [27,28,29,30]. Nevertheless, after the release of the sunflower whole genome sequence [4], a few studies have focused on specific multigenic families, such as WRKY, GSTs, and NACs, and on their expression during different environmental conditions [31,32,33]. In this work we analysed the expression of the HaLTP gene family in different organs and under different stimuli. We observed that sunflower LTP genes were mainly activated in the seeds, stamens, and roots, as already reported in other plant species, including barley and rice [11,12,34]. However, genes belonging to different HaLTP groups were far more expressed in some organs than in others, such as Group 1 genes in the stamen and Group 3 genes in the roots (Figure 5).

Group 2 HaLTP genes showed high expression values exclusively in the seed, with no response to any of the treatments mimicking stress. The presence of LTPs in the seed was observed in several plant species, such as maize [35], wheat [36], and sunflower, in which a specific role has been hypothesised for the fast mobilisation of lipids during germination [37]. Group 2 HaLTPs had a major similarity with the 2S seed storage proteins. The 2S seed storage family belongs to the prolamin superfamily, which includes LTPs. Both LTPs and 2S families share a conserved motif of cysteine sequences and are required for seed development and maturation [38,39]. 

Regarding treatments mimicking biotic and abiotic stress in the root, we detected several HaLTP genes that were specifically regulated by such stress. In particular, the ABA cDNA library showed the greatest number of overexpressed HaLTP genes belonging to Groups 1, 3, and 4 (Figure 6). These data also reflect the distribution of ABA cis-regulatory elements in the upstream nucleotide sequences of LTP genes of sunflower (Figure 4). Abscisic acid treatment rapidly induces the overexpression of non-specific LTP genes of types 1 and 2 in bromegrass, which is likely due to the activation of a signalling cascade [40]. Furthermore, similar activation of LTP genes in the same classes under ABA stimuli has been detected in rice roots [40,41]. In addition to ABA, IAA treatment also determined the extensive regulation of HaLTP genes belonging to Groups 1, 3, and 4 (Figure 6), which are also the groups of genes in which IAA cis-promoting elements are more frequent (Figure 4). Indole acetic acid is a plant hormone of the auxin class, and stimulation by auxin can largely affect the expression of LTP genes in bean roots. This increased expression might be correlated to a role in cortical tissue development [42]. 

Concerning MeJA and SA libraries, we detected the regulation of Group 1 and 4 HaLTP genes, especially through overexpression, except for one gene belonging to Group 1 that showed a negative fold change (Figure 6). Previous studies in other species reported activation of LTP gene expression by both stimuli in various plant organs, indicating a possible role of LTPs in plant stress resistance and defence to the environment [43,44].

Finally, sunflower roots exposed to salt stress showed major overexpression of HaLTP genes belonging to Group 4. It has been demonstrated that NaCl treatment upregulates the expression of LTP genes in the roots of *Arabidopsis thaliana*, and this activation seems to play a pivotal role in preventing osmotic stress and reducing susceptibility to a saline environment [45].

Group 3 HaLTPs show high similarity to the DIR1 LTPs of other species. The DIR1 proteins are regulated in response to the systemic acquired resistance (SAR) pathway in Arabidopsis [15]; many genes that are activated by SAR are also regulated in response to abiotic stimuli, such as ABA, IAA, MeJA, and SA [46]. 

Group 4 HaLTPs, which show sequence similarity with GPI-anchored LTPs, are not generally identified in the literature as possible targets during abiotic environmental changes, which contrasts with what was observed in our analysis; nevertheless, some GPI-LTPs have been described as activated by cold stress in Arabidopsis [47]. Interestingly, no response to any of the treatments mimicking stress was observed for Group 2 HaLTP genes, which are activated uniquely in the seed (Figure 5).

In conclusion, considering the expression analysis in the root and consensus motifs detected in HaLTPs genes (Figure 2, Figure 5 and Figure 6), we can hypothesise that Motifs 1, 2, and 3, which are specific to Group 1, are typical of LTPs involved in the response to ABA and IAA, whereas Motif2, associated with Motif4 (Group 4), is specific of LTPs activated under several abiotic stimuli, such as ABA, ACC, NaCl, SA, and KIN. Notably, Motif4, which is distinctive of Group 3, is present in LTPs that seem significantly regulated by a few biological stimuli, including ABA and IAA treatments (Figure 1 and Figure 6). 

The LTPs belonging to Group 2 was characterised by a specific protein motif (Motif5, Figure 1, Appendix A), and it can be hypothesised that Group 2 LTPs are specifically involved in transferring lipids in the seed, a role already reported for some LTPs of other plant species [18]. Once this hypothesis is confirmed at the biological level, Group 2 HaLTP-encoding genes could be of relevant interest for breeding in an important oilseed crop, such as sunflower.

This study represents the first genome-wide identification and characterisation of the LTP multigenic family in a plant species, the sunflower. We performed a detailed structural analysis of the LTP gene family, which led to the identification of four groups of candidate LTPs genes. Functional genomic data suggested that the different groups of HaLTP genes have specific expression patterns, depending on cis-regulatory elements in the promoter, and possible different functions due to protein structure. Group 2 HaLTP genes were apparently regulated specifically in seed, whereas Groups 1, 3, and 4 were more prone to be activated in different organs and in response to biotic or abiotic stress.

## 4. Materials and Methods

### 4.1. Retrieval of LTP Sequences in the Sunflower Genome

The *H. annuus* v. 2.0 genome, proteome, and annotation files were downloaded from the National Center for Biotechnology Information (NCBI). Putative LTP sequences were retrieved through a BLASTP homology search [48] against known protein sequences of sunflower LTPs reported in a previous study [23]. The parameters for the homology search were set to a sequence similarity of 60% and an E-value of 10E-3. Candidates were identified as *H. annuus* LTP (HaLTP).

Sequences identified by the Blast search were further investigated with InterProScan [49] to find common domains. The PF00234 Stockholm alignment was downloaded from Pfam and supplied to HMMER [26] to expand the LTP selection to possible divergent protein sequences. HMMER was used with default parameters, selecting full sequences with the best domain E-value below 10E-3. Sequences that had obvious errors in the description were discarded.

### 4.2. Sequence Alignments and Phylogenetic Analysis

Candidate HaLTP protein sequences were aligned by using Muscle on MEGA X software with default options [50]. Alignments were then used to construct a phylogenetic tree using the maximum likelihood parameter on MEGA X with a JTT model and 1000 bootstrap replications.

### 4.3. Structural Characterisation of HaLTP Sequences

The Gene Structure Display Server (GSDS) was exploited to retrieve the exon–intron structure of HaLTP gene sequences; gene and CDS sequences in FASTA format and the phylogenetic tree in Newick format were supplied to the software.

Structural motifs in HaLTP protein sequences were discovered using the web version of Multiple EM for Motif Elicitation (MEME) [51]; default parameters were used except for the number of motifs, which was set to 5. Structural motif domains were visualised using both MEME and motif scanning algorithm options supported by the MEME website.

Protein sequences of putative HaLTPs were submitted to SignalP 5.0 using default parameters to remove possible peptide signals [52]. Subsequently, mature proteins were supplied to EMBOSS-Pepstats [53] to calculate the isoelectric point (pH) and molecular weight (Da) using default options. Differences in molecular weight and isoelectric point between groups were assessed with analysis of variance (ANOVA) and post hoc Tukey tests with a significance threshold of 0.05.

The sequence similarity of putative HaLTPs was analysed using BLASTp on Swiss-Prot, a manually annotated and reviewed database, with default setting. Proteins similar to HaLTPs were further investigated by searching functional domains on the Pfam database using InterProScan [49].

### 4.4. Analysis of Chromosomal Distribution, Gene Duplication Events, and Putative Pseudogenes

Lipid transfer proteins gene locations in sunflower chromosomes were obtained using the annotation file provided by NCBI, and LTP gene density was visualised using the “Rideogram” R package [54].

HaLTP duplication events in the sunflower genome were analysed using MCScanX [55]. The input annotation and protein sequence similarity files were generated using BLASTP and GFF annotation files of sunflower, as suggested by the manual. 

HaLTPs genes, which were located within the frame of the 200 kbp, were considered by MCScanX as duplicated in tandem, whereas MCScanX algorithm detected whole genome duplication (WGD) events evaluating genes’ similarity and different genomic position in order to create anchored loci, which are used to estimate collinearity region between chromosomes.

Information about HaLTP paralogues was obtained using perl scripts provided by MCScanX software.

The ratio between non-synonymous and synonymous substitutions (Kn/Ks) in the pairwise comparison of HaLTP protein sequences was measured using both PAL2NAL [56] and SNAP [57]. Clustal Omega was used to generate the alignment for PAL2NAL, as suggested by software.

The occurrence of putative pseudogenes among candidate HaLTP sequences was searched by supplying sunflower transcript sequences to Augustus [58] using *Arabidopsis* as a predicting species; *H. annuus* lipid transfer protein sequences showing altered frameshifts, missing start and/or stop codons, and truncated translation were flagged as putative pseudogenes.

### 4.5. Expression Analysis and Identification of Putative Cis-Regulatory Elements

Expression of HaLTP sequences was analysed in available cDNA libraries from different sunflower organs and from roots exposed to treatments mimicking stress. Illumina libraries were downloaded from the Sequence Read Archives (SRA). cDNA obtained after treatments mimicking biotic/abiotic stress (abscisic acid, ethylene, brassinosteroids, gibberellic acid, auxin, kinetin, methyl jasmonate, salt, PEG, salicylic acid, and strigolactones) are available under project accession SRP092742. These cDNA libraries were obtained as described by Badouin et al. [4]; libraries of *H. annuus* (cultivar SK02R) organs derived from ligule, leaf, root, seed, pistil, and stamen are available under accession PRJNA483306.

Reads from cDNA libraries were trimmed using Trimmomatic v0.38 [59], removing low-quality reads and adapters. Then, high-quality reads were mapped on the sunflower transcriptome using the CLC genomics workbench (v9.5.3, CLC-BIO, Aarhus, Denmark) with the following parameters: mismatch penalties = 2, gap open penalties = 3, length fraction = 0.9, and similarity fraction = 0.9. Reads mapped for each gene were normalised with reads per kilobase of exon per million mapped reads (RPKM) [60].

Regarding libraries from roots exposed to treatments mimicking biotic and abiotic stress, differentially expressed genes (DEGs) were retrieved by comparing treatments versus respective control libraries. DEGs were obtained using EdgeR [61] with a quasi-likelihood statistical test. Genes with an absolute fold change value over 2 and false discovery rate (FDR)-corrected p-value under 0.05 [62] were considered differentially expressed. Transcripts that showed an RPKM value < 1 in each library were not considered for differential expression analysis.

To retrieve cis-acting regulatory elements, nucleotide sequences in the frame of the 1500 bp upstream of the putative HaLTP coding sequences were analysed using PlantCare [63].

## Figures and Tables

**Figure 1 plants-11-00664-f001:**
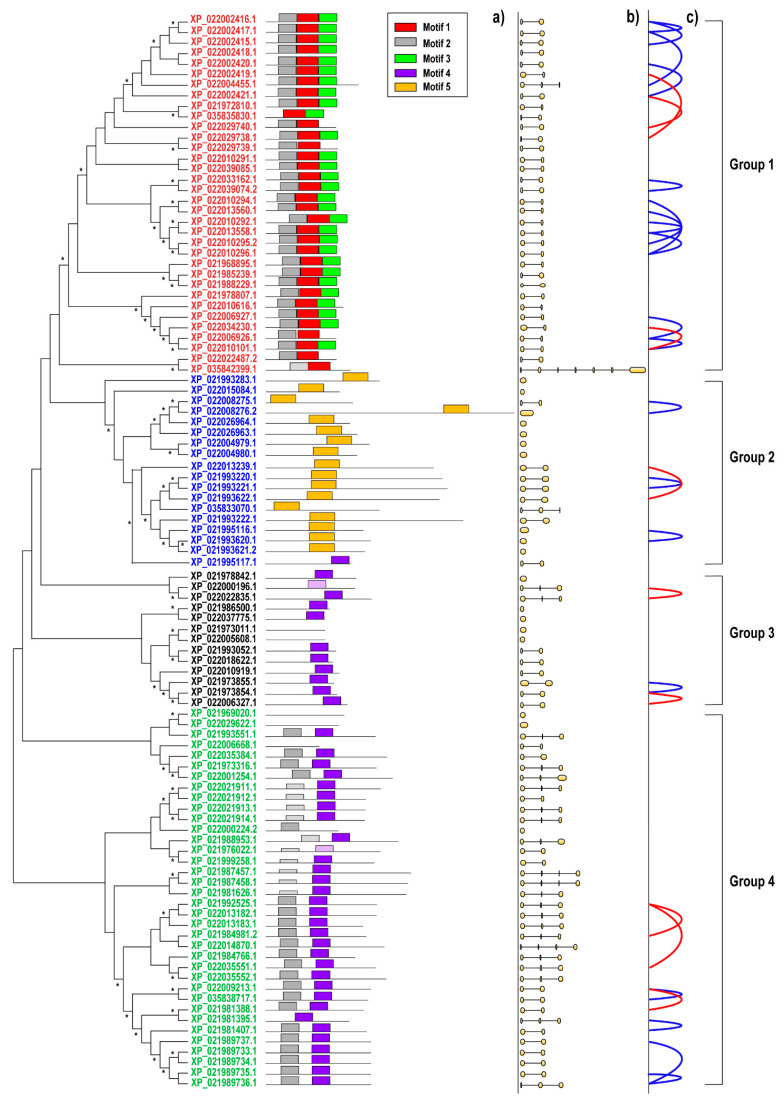
Phylogenetic analysis by maximum likelihood of the lipid transfer protein (LTP) family in *Helianthus. annuus*. Proteins were split into groups following phylogenetic and structural analysis: Group 1 (red font), Group 2 (blue font), Group 3 (black font), and Group 4 (green font). The asterisks at each node indicate bootstrap values > 50%. For each sequence, (**a**) the structural organisation, based on the occurrence of specific protein motifs; (**b**) on the expected exon–intron structure of the corresponding gene (CDSs are indicated with yellow circles or ellipses, introns with black lines); (**c**) the occurrence of genes putatively originating by tandem duplication (blue arch) or by whole genome duplication events (red arch) are reported. MEME consensus motifs are as follows: Motif1 (Red, CCNGVKGLNAAAKTTADRQAACGCLKSAYSSJSGI), Motif2 (Grey, YAEASTCGQVLSSLSPCLNYLTGGGSVPP), Motif3 (Green, AGNAASLPGKCGVNIPYKISPSTDCSKVQ), Motif4 (Purple, NGGGSSLGLNINQTLALELPKACNVQTPP), and Motif5 (Yellow, QKEZQLLQQCCQZLQNVEEQCQCEAVKQVFRZAQQQVQQQQ). Motifs in a pale colour are similar to Motif2 and Motif4, but lowly conserved and identified with a motif scanning algorithm.

**Figure 2 plants-11-00664-f002:**
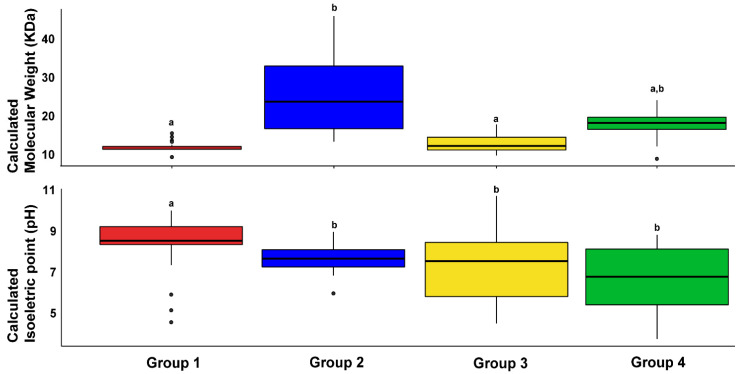
Box plots of calculated molecular weight (KDa) and calculated isoelectric point (pH) for each lipid transfer protein (LTP) group in the *Helianthus annuus* genome. Statistical differences among groups were detected by ANOVA and Tukey’s test (*p*-value < 0.05). Black lines represent the mean value for each group. Boxes include 50% of the values for each group. Letters “a” and “b” indicate significant differences between groups.

**Figure 3 plants-11-00664-f003:**
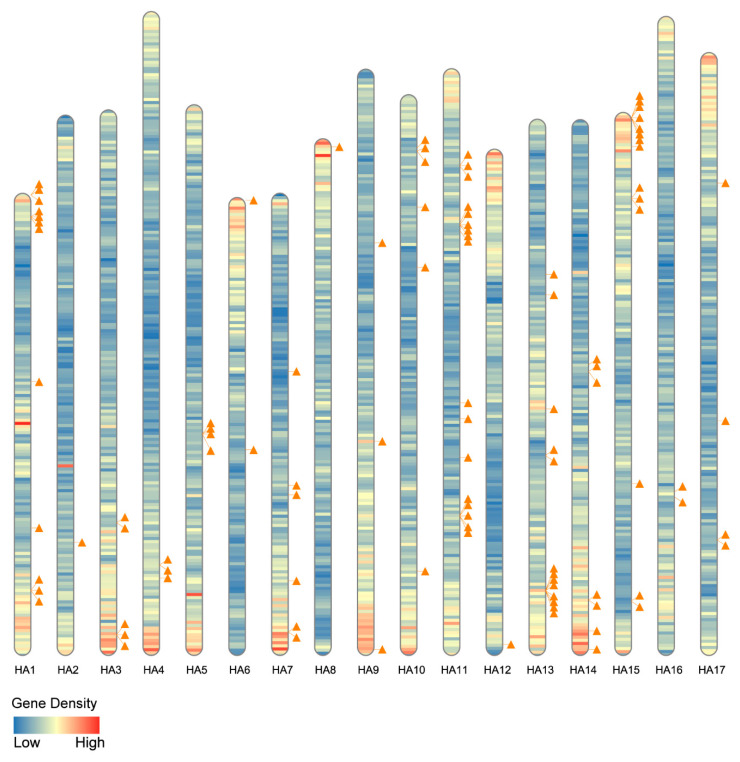
Lipid transfer protein (LTP) family distribution in the *Helianthus annuus* genome. Segments represent the whole chromosome complement of sunflower (*n* = 17); colours from blue to red show the gene density per chromosome. Orange triangles indicate HaLTP locations on the chromosomes.

**Figure 4 plants-11-00664-f004:**
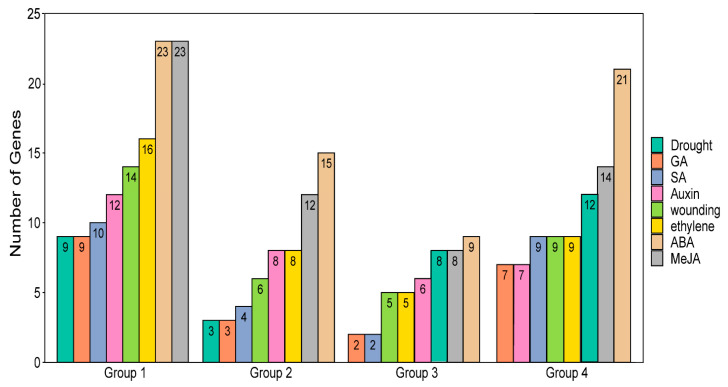
Number of lipid transfer protein (LTP) genes per Group (1–4) with at least one cis-acting element in their promoters responsive to abscisic acid (ABA), methyl jasmonate (MeJA), ethylene, wounding, auxin, drought, salicylic acid (SA), and gibberellic acid (GA).

**Figure 5 plants-11-00664-f005:**
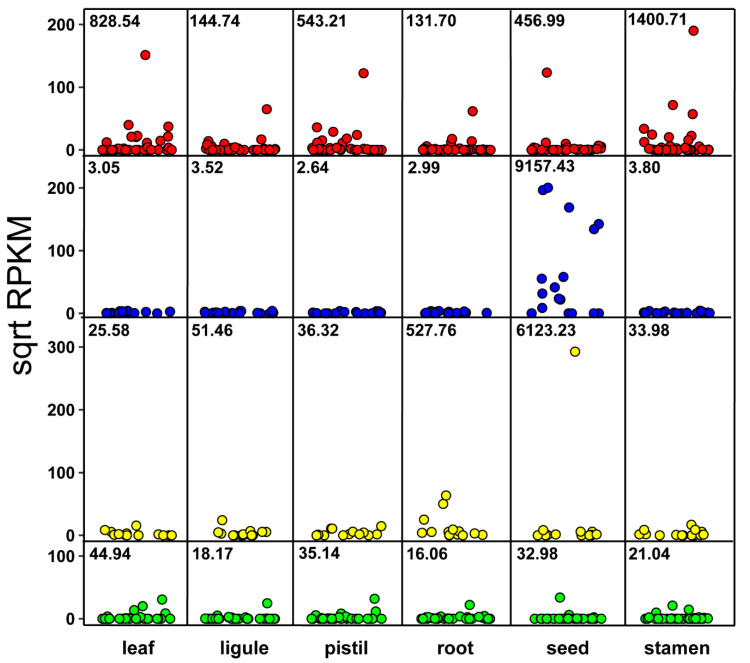
Expression values (shown as the square root of RPKM) of each lipid transfer protein (LTP) group in six organs of *Helianthus annuus*. Red dots = Group 1, blue dots = Group 2, yellow dots = Group 3, green dots = Group 4. In each box, the average RPKM value was reported.

**Figure 6 plants-11-00664-f006:**
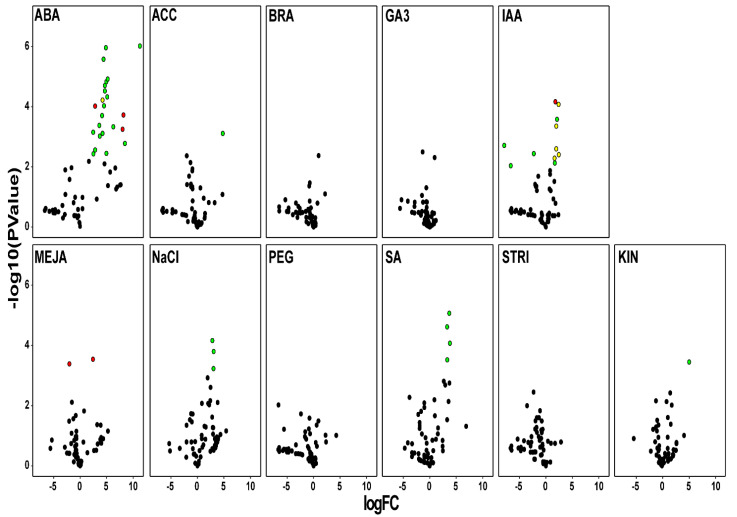
Volcano plots showing genes differentially expressed in roots of plantlets treated with different substances (ABA: abscisic acid; MeJA: methyl jasmonate; ACC: ethylene; BRA: brassinosteroids; GA3: gibberellic acid; IAA: auxin; KIN: kinetin; PEG: polyethylene glycol; SA: salicylic acid; STRI: strigolactones) for each lipid transfer protein (LTP) group after treatment. Red dots = Group 1 HaLTP significantly regulated; yellow dots = Group 3 HaLTP significantly regulated; green dots = Group 4 HaLTP significantly regulated; black dots = HaLTP not significantly regulated. FC = fold change.

## Data Availability

The data presented in this study are openly available in Sequence Read Archive (SRA) at BioProject accession SRP092742 and PRJNA483306.

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
