# Peer review of "In Silico Genome-Wide Characterisation of the Lipid Transfer Protein Multigenic Family in Sunflower (H. annuus L.)"

_plants, 2022, doi:10.3390/plants11050664_

Round 1
Reviewer 1 Report
Using a plant species as a model, this study is the first effort to identify and characterise the LTP multigenic family on a genome-wide scale. The manuscript is well written and it can be published with minor revision.
The abstract must be rewritten with more detailed information in terms of results.
Please start each sentence with full form, no matter you are using abbreviations.
Even though the authors provide an extensive literature
review of the Introduction, they should enhance and reconstruct it in the introduction section. They should provide the scope of the paper clearly.
In general, there is a repetition of information that might have been omitted.
Check the English language of the results and discussion section.
Author Response
We agree with reviewer 1, the abstract has been fully revised and more results have been added. The introduction has been modified, with a deeper description of the literature cited. Moreover, the scope of the work has been rewritten. English language was revised by a native scientist who has checked the text again.
Reviewer 2 Report
Comments and suggestions as an attached word document

Author Response
All suggestions by reviewer 2 have been accepted and the manuscript was changed accordingly. Concerning line 581 we mentioned a reference by a book which has been formatted according to Plants editorial guideline.
Reviewer 3 Report
Dear authors
This manuscript regarding Gene family identification of lipid transfer protein gene and expression analysis in sunflower is well written. This manuscript might be a good contribution for researchers working in this plant. Thus, I would like to recommend this manuscript for publication after a minor revision. My few is given comments below:
Since all data of this manuscript are analyzed In silico,
Thus, the Authors should perform the real-time PCR for at least some selected genes from the phylogenetic tree, representing each subclade of the phylogenetic tree. Thus, they can validate the data from Sequence Read Archives from real-time PCR expression data.
If the above is not possible from your side, kindly change your title to “In Silico Genome-wide characterization…”
Kindly check the reference format in the text according to journal format style.
Author Response
We would like to thank Reviewer 3 for the suggestion. We decided to change the title as suggested by the reviewer.
Reviewer 4 Report
The manuscript describes a genome wide characterization of the lipid transfer protein family in sunflower. The study is interesting and is well written.
Some minor suggestions are made to improve the manuscript.
Line 116 – Is that “Motif2 and Motif4” or Motif2b and Motif4b?
Line 291 – Identified differentially expressed HaLTP genes in roots, from Groups 1, 3, and 4, treated with distinct hormones and compounds, should be identified and described in a supplementary Table.
Please revise the scientific names in the "references" section, the Arabidopsis name in reference number 15 and the scientific name in reference 18 need amendments.
Author Response
We agree with reviewer 4. Manuscript has been corrected accordingly with suggestions and a table of differentially expressed HaLTPs in roots has been added to supplementary material. Concerning line 116 motif 2b and motif 4b correspond to motif 2 and 4 and have not been distinguished from motifs 2 and 4 in the text. In fact, are the same motif retrieved with another MEME algorithm as described in material and methods. We decided to eliminate the reference to motif 2b and 4b in the text.